# Immunotherapy for SMARCB1-Deficient Sarcomas: Current Evidence and Future Developments

**DOI:** 10.3390/biomedicines10030650

**Published:** 2022-03-11

**Authors:** Carine Ngo, Sophie Postel-Vinay

**Affiliations:** 1ATIP-Avenir Group, Inserm Unit U981, Gustave Roussy, 94800 Villejuif, France; carine.ngo@gustaveroussy.fr; 2Department of Pathology, Gustave Roussy, 94800 Villejuif, France; 3Drug Development Department, DITEP, Gustave Roussy, 94800 Villejuif, France

**Keywords:** SMARCB1, immunotherapy, immune checkpoint inhibitor, epithelioid sarcoma, rhabdoid tumor, SWI/SNF

## Abstract

Mutations in subunits of the SWItch Sucrose Non-Fermentable (SWI/SNF) complex occur in 20% of all human tumors. Among these, the core subunit SMARCB1 is the most frequently mutated, and SMARCB1 loss represents a founder driver event in several malignancies, such as malignant rhabdoid tumors (MRT), epithelioid sarcoma, poorly differentiated chordoma, and renal medullary carcinoma (RMC). Intriguingly, SMARCB1-deficient pediatric MRT and RMC have recently been reported to be immunogenic, despite their very simple genome and low tumor mutational burden. Responses to immune checkpoint inhibitors have further been reported in some SMARCB1-deficient diseases. Here, we will review the preclinical data and clinical data that suggest that immunotherapy, including immune checkpoint inhibitors, may represent a promising therapeutic strategy for SMARCB1-defective tumors. We notably discuss the heterogeneity that exists among the spectrum of malignancies driven by SMARCB1-loss, and highlight challenges that are at stake for developing a personalized immunotherapy for these tumors, notably using molecular profiling of the tumor and of its microenvironment.

## 1. Introduction

Sarcomas represent a very heterogeneous group of rare soft tissue and bone cancers, which comprise more than 100 subtypes, and arise in all age groups, making it particularly difficult to treat. Sarcomas have traditionally been classified according to their histopathological aspect, together with some specific immunohistochemical markers [1]. The advent of next generation sequencing techniques (notably whole exome and RNA sequencing), has allowed to build a new molecular classification of sarcomas, notably based on mutations, copy number alterations, and gene fusions [2]. More recently, other classifiers have emerged, notably based on DNA methylation profiles [3], or on digital pathology associated with artificial intelligence/deep learning analyses, which all represents valuable additional tools for sarcoma classification [4].

Beyond genetic abnormalities, several sarcoma subtypes are driven, at least in part, by epigenetic dysregulation. Notably, alterations in subunits of the mammalian SWItch Sucrose Non-Fermentable (mSWI/SNF) chromatin remodeling complex have been found in 20% of human cancers and are frequent in certain sarcoma subtypes [5]. SWI/SNF plays essential roles in chromatin organization, genome maintenance, and gene regulation [6]. Specifically, the core subunit SMARCB1 (INI1), is the most frequently inactivated subunit in mesenchymal neoplasms [7], with SMARCB1 deficiency being characteristic of malignant rhabdoid tumors (MRT), epithelioid sarcoma (ES), and most of poorly differentiated chordoma.

Despite the tremendous heterogeneity of sarcomas and the impressive progresses made in understanding the disease biology thanks to the molecular classification, treatment still most often relies on “one size fits all” therapeutic strategy, based on poly-chemotherapy, surgery, and radiotherapy, which brings little benefit at the cost of high toxicity. Over the past ten years, cancer immunotherapy particularly with immune checkpoint inhibitors (ICI) targeting the Programmed Death-1 (PD-1)/ligand-1 (PD-L1) axis, has revolutionized the outcome of several aggressive cancers, such as metastatic melanoma, non-small-cell lung cancer (NSCLC) or renal cell carcinoma [8]. In sarcoma, development of immunotherapy has encountered challenges due to the rarity and high heterogeneity of the disease. Nonetheless, recent scientific advances and clinical results have enabled the identification of sarcoma subgroups that may most benefit from ICI, together with potential predictive biomarkers of efficacy. Notably, mSWI/SNF defects emerged recently as a putative promising predictive biomarker of sensitivity to ICI [9,10]. 

This review will discuss the current evidence for using immunotherapy in SMARCB1-deficient sarcoma.

## 2. SMARCB1

SMARCB1 (aka SNF5, INI1 and BAF47), is one of SWI/SNF core subunits. mSWI/SNF remodeling complexes, also known as BRG1/BRM-associated factor (BAF) complexes, exist in three forms: the canonical BAF (cBAF), Poly-Bromo BAF (PBAF), and non-canonical BAF (ncBAF), which differ in subunit composition and patterns of genomic targeting [6]. Each complex is composed of three or five core subunits (SMARCC1, SMARCC2, SMARCD1-3, SMARCB1, and SMARCE1 for cBAF and PBAF; SMARCC1, SMARCC2 and SMARCD1-3 for ncBAF), one ATPase subunit (SMARCA2 or SMARCA4 for cBAF and PBAF), some complex-specific subunits, and several additional regulatory subunits [6,11]. The mSWI/SNF complexes are involved in multiple important cellular processes, including cell differentiation and cell proliferation [12], or transcription by facilitating the binding of the transcription complex machinery and specific transcription factors to target genes [6]. 

### 2.1. SMARCB1 Structure and Functions

The *SMARCB1* gene, located at 22q11.23, encodes a 47 kDa protein of four functional domains with frequent loss of function mutations in cancer [13] (Figure 1). Interestingly, SMARCB1 was first identified as a binding partner of the human immunodeficiency virus-type 1 (HIV-1) integrase (to which it binds through its two highly conserved imperfect repeat domains, Rpt1 and Rpt2), and was, thereafter, named integrase interactor 1, INI1 [14,15]. Through its Rpt1 domain, SMARCB1 further interacts with the SWIRM domain of SMARCC1 to form the BAF and PBAF forms of mammalian SWI/SNF complexes [16]. Very recently, this N-terminal region, which is notably mutated in rhabdoid tumors, was further found to undergo a coil-to-helix transition upon binding of SMARCC1, revealing a unique binding interface [17]. SMARCB1 also contains a N-terminal winged helix DNA-binding domain, which is regularly mutated in schwannomatosis [18]. The highly conserved putative coiled-coil C-terminal α helix domain (CTD)—also frequently altered in cancer—was found to mediate chromatin remodeling through interaction with the acidic patch of the nucleosome [19]. 

SMARCB1 plays a key role in maintaining SWI/SNF integrity [11], which is required for its targeting to regulatory regions where it opposes Polycomb-mediated repression at bivalent promotors [20,21]. SMARCB1 also commonly acts as a tumor suppressor by transcriptionally regulating the cell cycle, proliferation, and differentiation. For example, SMARCB1 has been shown to regulate the activation of CyclinD1/CDK4 signaling, to repress RB target genes, and to participate to the regulation of c-MYC-associated transcriptional programs [22].

### 2.2. SMARCB1 Alteration in Cancers

The mSWI/SNF complexes were first linked to cancer in 1998 after O. Delattre and colleagues identified *SMARCB1* biallelic inactivation as a driver of MRT [23]. Subsequently, complete loss of SMARCB1 expression has been found in a variety of tumors (Table 1), including malignancies of the central nervous system, soft tissue, kidney, sinonasal and gastrointestinal tract [24]. Here, we will focus on SMARCB1-deficient sarcomas.

Malignant rhabdoid tumor (MRT) are a rare, highly aggressive tumor that predominantly affects infants and young children below 3 years of age. They may affect the kidney (Rhabdoid Tumor of the Kidney, RTK), central nervous system (Atypical Teratoid Rhabdoid Tumor, ATRT) and extrarenal sites (Extra-Renal Rhabdoid Tumor, ERRT). They present a remarkably simple genome and are uniquely characterized by a biallelic deletion of *SMARCB1* in more than 95% of cases [25,26,27]. In mice, whereas homozygous inactivation leads to early embryonic lethality, heterozygous loss promotes the development of undifferentiated or poorly differentiated sarcomas consistent with MRT observed in infants [28,29,30], highlighting the central role of *SMARCB1* in this disease.

Epithelioid sarcoma (ES) affects patients over a wide range of ages [1]. It is classically classified in two clinicopathological subtypes: (i) the distal (or classical) type, which predominantly arises from the superficial distal sites and is most prevalent in adolescents and young adults, is characterized by nodules of epithelioid to spindle cells with central necrosis, surrounded by chronic inflammatory infiltrate; and (ii) the proximal type which arises in deep central truncal sites in older patients, are characterized by sheet-like growth of large, sometimes pleomorphic epithelioid tumor cells, with a poorer prognosis, and a tendency to local recurrence and rapid metastasis. Rhabdoid cells may be seen in both histological subtypes and some ES may demonstrate hybrid histological features, highlighting the limitation of such classification. Median survival is ~52 weeks in patients with metastasis [31]. Complete loss of SMARCB1 expression is found in more than 90% of both subtypes [32]. Despite having many features overlapping those of ERRT, there are fundamental differences in clinical behavior, genomics, and presumably oncogenic mechanisms between these tumor types. First, the mechanisms leading to SMARCB1 inactivation are more diverse in ES than in MRT, and are still not fully understood. Genetic inactivation of *SMARCB1* have been reported to occur in 10% to 83% of cases [33,34,35,36], mostly in the form of homozygous deletions and less frequently monoallelic deletion or more rarely as nonsense, frame, or deleterious point mutations [34,35,37]. Epigenetic mechanisms, notably following SMARCB1 mRNA downregulation by microRNAs [38,39,40], have also been described. Unlike MRT, ES presents a moderate to high tumor mutation burden [36]. Further, the sometimes co-occurring loss of CDKN2A suggests that oncogenic mechanisms are different than those reported in MRT—where SMARCB1 loss leads to cell cycle deregulation through CDKN2A [36,41]. Altogether, this suggests that, contrary to MRT, additional genomic abnormalities cooperate with SMARCB1 loss to drive ES. Treatment of localized ES relies on surgery, most often subsequent to neoadjuvant chemotherapy and/or followed by adjuvant radiation therapy [42]. Unfortunately, relapse or local recurrence frequently occur, especially for proximal ES, with nodal or metastatic involvement. In 2020, tazemetostat (Tazverik^®^, Epizyme) was the first ever epigenetic therapy to be approved in solid tumors. Tazemetostat was registered for the treatment of advanced SMARCB1-defective ES. Still, only 15% of patients respond to EZH2 inhibitors [43] and resistance rapidly develops, urgently calling for complementary therapeutic approaches. 

Poorly differentiated chordoma is an extremely rare tumor which mostly arises in the skull base in young children, and more rarely in the sacrococcygeal area. It is characterized by nests of epithelioid cells with notochordal differentiation and focal rhabdoid morphology, and by the loss of SMARCB1 expression virtually in all cases [44]. Cytogenetic studies have identified frequent homozygous deletions of *SMARCB1* [45,46,47,48]. Poorly differentiated chordoma is associated with a worse prognosis than that of conventional chordoma [44,49]. Treatment consists of a combination of surgery, radiation therapy, and chemotherapy.

Epithelioid malignant peripheral nerve sheath tumor, accounting for less than 5% of malignant peripheral nerve sheath tumor (MPNST), differs from conventional MPNST by the strong and diffuse expression of S-100 protein and SOX10, a rare association with type 1 neurofibromatosis, occasional origin in a schwannoma, and loss of expression of SMARCB1 following homozygous deletion, nonsense, frameshift, or splice site mutations in up to 75% of cases [50,51]. 

Other SMARCB1-negative sarcomas. Loss of SMARCB1 expression has also been observed in up to 40% of myoepithelial carcinoma, notably following homozygous deletion [52,53], and in 17% of extra-skeletal myxoid chondrosarcoma [54]. Finally, synovial sarcoma (SS) also shows reduced SMARCB1 function, following its eviction from the SWI/SNF complex subsequent to the incorporation of the hallmark SS18–SSX oncoprotein [55]. Because this causes an indirect SMARCB1 functional defect rather than a direct loss of the protein expression, we will not discuss SS here.

## 3. SMARCB1 Deficiency and Anti-Tumor Immunity

The immune system plays a key role in controlling tumor development and metastasis. The tumor immune microenvironment (TME) is traditionally classified into three phenotypes: immune-inflamed, immune-excluded, and immune-desert [56]. Immune checkpoint inhibitors (ICI), notably those targeting the PD-1/-L1 or CTLA-4/CD28 checkpoints, have revolutionized the prognosis of several aggressive diseases [57]. ICI block signals that inhibit CD8+ T-cell activation, thereby reinvigorating in situ anti-tumor responses. The immune-inflamed profile, characterized by CD8+ and CD4+ tumor-infiltrating lymphocytes (TILs), and PD-L1 expression on immune and tumor cells, is classically associated with better response to ICI [58,59]. Here, we will present the current knowledge on the TME-related factors and tumor-related factors, which support a potential role for immunotherapy in SMARCB1-defective tumors, with a focus on ICI. 

### 3.1. SMARCB1 Deficiency and Tumor Cell Immunogenicity

Several tumor-related factors have been shown to influence response to immune therapies, including the tumor mutational burden (TMB) and neoantigen load [60], or the mutation in genes that influence tumor ability to escape immune surveillance [56] (e.g., MYC or RAS [61,62]) or respond to interferon-gamma (IFNγ) signaling [63]. Interestingly, recent data have reported a link between deficiencies of certain mSWI/SNF subunits and increased mutational load or enhanced interferon response [6,64,65]. 

SMARCB1 modulates tumor cell immunogenicity through various mechanisms. Using immunocompetent mouse models and human tumors, Leruste and colleagues showed that, despite their very simple genome and extremely low mutational burden, MRT are immunogenic [66]. Mechanistically, SMARCB1 deficiency favored the de-repression of multiple endogenous retroviral elements (ERVs), leading to cytosolic double-stranded RNA (dsRNA) accumulation which activates cytoplasmic sensors, such as TLR3 and MDA5, and subsequent cell-autonomous IFN-α and IFN -λ responses (Figure 2). Interestingly, recent studies have also reported that de-repressed ERVs could generate tumor specific antigens presented at the tumor cell surface by MHC-I molecules and recognized by specific T-cells [67,68]. An increased anti-tumor innate immune response driven by ERVs de-repression has recently been observed in multiple tumor types [69,70], and correlated to responsiveness to anti-PD-L1 therapy [71]. 

Chun et al. further discovered that MRT and the ATRT-MYC subgroups presented a immunologically active microenvironment, with increased cytotoxic T-cell infiltration [72]. Interestingly, they did not evidence ERV de-repression, but identified nine aberrantly expressed tumor antigen genes, whose expression was specific to MRT and associated with increased T-cell infiltration. 

In RMC, Msaouel et al. identified that the dsDNA–sensing cGAS/STING pathway was activated and associated with enhanced tumor immunogenicity (Figure 2) [73]. They suggested that SMARCB1 loss resulted in enhanced replication stress following the activation of the c-MYC pathway, and subsequent increased cell-cycle checkpoint activation and DNA damage. However, how exactly SMARCB1 loss leads to cGAS/STING activation, and whether this is also the case in other SMARCB1-defective carcinomas or sarcomas, remains to be elucidated.

### 3.2. SMARCB1-Deficient Tumors’ Immune Microenvironment

#### 3.2.1. Tumor-Infiltrating Lymphocytes (TILs)

The tumor immune microenvironment of pediatric cancers has traditionally been anticipated to be poorly infiltrated or immune-excluded [74], notably as a consequence of their simple genome and low TMB. Surprisingly, despite sharing the same unique genomic alteration, rhabdoid tumors are molecularly heterogeneous [75] at the epigenomic and transcriptomic level. 

By studying the DNA methylation, Chun and colleagues identified five MRT subgroups: (1) ATRT-MYC, (2) ATRT-TYR, (3) RTK, (4) ERRT, and (5) ATRT-SHH [72]. Groups 1, 3, and 4 presented a high expression of immune-related genes, notably those involved in antigen processing, antigen presentation, T-cell activation and homing, and innate immune response. This correlated with increased infiltration of CD8+ cytotoxic T-cells by immunohistochemistry and enriched immune-related gene signatures in RNA-Seq, alongside with increased infiltration of PD-L1+ CD68+ myeloid cells [72]. Similarly, Leruste and colleagues found that the ECRT and ATRT-MYC subgroups were highly infiltrated by CD8+ T-cells. Using single cell RNA and T-cell receptor sequencing, they identified clonally expanded resident memory CD8+ T-cells, suggesting the presence of a local tumor-specific anti-tumor immune response [66]. In mouse syngeneic models, MRT were highly infiltrated in PD-1+ TILs and were highly sensitive to anti-PD-1 therapy (75% of complete response and survival benefit). 

Despite its low TMB, RMC is also highly infiltrated by TILs, myeloid dendritic cells, neutrophils, and B-cells [73]. This overall suggests that, contrary to what was initially anticipated, certain subgroups of tumors with simple genomics are immunogenic and may benefit from immune therapy-based therapeutic approaches.

#### 3.2.2. Tertiary Lymphoid Structure (TLS)

TLS are heterogeneous aggregates of B- and T-cells, forming follicle-like structures containing a network of CD21+ and/or CD23+ follicular dendritic cells [76]. In breast, colorectal cancer, hepatocellular carcinoma, non-small cell lung cancer (NSCLC), and GIST, the presence of TLS was found to be associated with a favorable prognosis [77,78,79,80,81]. In NSCLC and melanoma, TLS-rich tumors are more infiltrated by CD4+ and CD8+ cells, supporting a role for TLS in modulating T-cells in the TME even outside TLS [82,83]. Interestingly, the presence and density of TLS correlate with therapeutic responses to ICI in several tumor types including frequent soft-tissue sarcomas (NCT02301039) [84], making it an additional important predictive biomarker [76]. The presence of TLS in SMARCB1-defective sarcoma and its association with response to immunotherapy have not been studied yet.

#### 3.2.3. Myeloid Populations

Beyond T-cells, many other immune cells affect tumor response to immune therapies. In particular, tumor-associated macrophages (TAMs) and myeloid-derived suppressor cells (MDSCs) have been associated with poorer response to ICI [85,86]. Despite their potential sensitivity to anti-PD-(L)1 therapy, MRT are also infiltrated by myeloid cells, including monocytes, macrophages with pro-tumoral M2-like signatures [66,72]. Recent studies have also reported a predominant infiltration of M2-like TAMs in ES and chordoma [87].

#### 3.2.4. Immune Checkpoint Inhibitors

Anti-PD-(L)1 therapy is the most frequently used ICI. High PD-L1 expression, using FDA-approved immunohistochemistry companion diagnostic tests (e.g., 22C3 pharmDx on Dako Omnis, Agilent, for pembrolizumab, MSD; or SP263 on Ventana for atezolizumab, Roche), is currently the most robust predictive biomarker of response to anti-PD-(L)1 agents, and is routinely used in patients with advanced NSCLC, bladder cancer, and head and neck squamous cell carcinoma [59,88,89,90,91,92]. 

MRT present a variable PD-L1 expression [93,94]. Abro and colleagues found that eight (out of 16) cases of ATRT/MRT were PD-L1-positive (tumor proportion score: 10–70%), and that nine cases displayed high PD-1 expression in TILs [93]. However, PD-1/-L1 expression was not predictive of survival. More recently, Forrest and colleagues investigated PD-L1 expression in 30 SMARCB1-negative sarcomas, including ES and anaplastic chordoma [94]. They found that 47% of cases were PD-L1 positive (≥1% of positivity in tumor cells or TILs), and that this was more frequent in extracranial sites, with 4/4, 4/9, 1/9, and 1/1 in ES, ERRT, ATRT, and anaplastic chordoma being PD-L1 positive. Using TCGA RNA-sequencing for analysis of tumor-infiltrating immune cells, they observed an inverse correlation between SMARCB1 mRNA levels and both CD8 and PD-L1 expression across multiple cancer types [94]. PD-L1 expression has also been explored in larger cohort of various sarcoma subtypes (including ES), with expression varying from 0 to 100% [94,95,96,97]. Interestingly, PD-1 seemed to be highly expressed when PD-L1 was absent [97]. In the study from Kim et al., where 7/7 ES were PD-L1 high (using a threshold of >10% of tumor cells for positivity), PD-L1 expression was significantly associated with shorter 5-year overall survival and an independent negative prognostic factor, thereby supporting its role as a potential therapeutic target. High PD-L1 levels have also been reported in chordoma, where it correlated with the presence of TILs, metastatic status, and worse prognosis [98,99]. 

Beyond PD-1/-L1, Leruste et al. reported that other immune checkpoint receptors (including HAVCR2/TIM-3 and LAG3) were expressed by CD8+ T-cells in extra-cranial RT and MYC-AT/RT, suggesting that T-cell exhaustion may contribute to immune escape in these diseases [66].

## 4. Clinical Efficacy of Immune Therapies in SMARCB1-Defective Tumors

Here, we will focus on the results of clinical trials evaluating the immune checkpoint inhibitors targeting the PD-1/PD-L1 and CTLA-4/CD80/86 checkpoints, as these are the only immune therapies evaluated in the clinic in these diseases so far.

### 4.1. Anti-PD(L)-1 Therapy as a Monotherapy

Clinical benefits of immune checkpoint inhibitor therapy in SMARCB1-negative sarcoma have been reported in several clinical trials and individual case reports (Table 2). Regarding clinical trials, five partial responses have been reported out of nine patients with MRT or ES treated with anti-PD(L)1 [100,101,102,103]. Three responses were long-lasting, with durations of 12, 13, and 18 months [100,101]. Forrest and colleagues further reported some clinical benefit of anti-PD-(L)1 therapies in three pediatric patients with SMARCB1-negative diseases [94]. The first patient, who was treated as first-line therapy for an ES (PD-L1+ on 40% of the tumor cells), presented a prolonged stable disease and stayed 12 months on therapy; the second patient with PD-L1+ (5% of tumor cells) anaplastic chordoma presented a shrinkage of some lesions and overall stable disease for 9 months on nivolumab; the last patient received pembrolizumab as first-line treatment for a MRT and presented a stable disease during 15 weeks. TLS were not evaluated in these cases.

### 4.2. Anti-PD-(L)1 Therapy in Combination

Several combinatorial strategies have been proposed in order to improve the efficacy of ICI and fully unleash their antitumor potential (Table 3). These include combinations with other ICI, targeted therapies, anti-angiogenic agents or epigenetic drugs which all aim at synergistically boosting the anti-tumor immune response [6,104]. 

#### 4.2.1. Dual ICI Combination

Combined targeting of the anti-PD-1 nivolumab with or without the anti-CTLA4 ipilimumab in metastatic sarcoma has been first evaluated in the randomized clinical trial Alliance A091401 (NCT02500797) [105]. Across all sarcoma subtypes, the combined administration resulted in an objective response rate of 16% (6/38 patients) and median progression-free survival of 4.1 months, versus 5% (2/38 patients), and 1.7 months with nivolumab monotherapy; the only patient with ES enrolled in this trial received nivolumab monotherapy and did not respond to treatment. By contrast, Pecora and colleagues reported a rapid and complete response to nivolumab and ipilimumab in a chemotherapy- and tazemetostat-pretreated patient with advanced SMARCB1-negative ES [106], calling for further molecular investigation of the determinants of response to ICI in this context. The nivolumab plus ipilimumab combination is currently being further evaluated in metastatic sarcoma of rare subtype including ES and chordoma (NCT04741438) (Table 4). Additionally, based on the preclinical evidence that supports the presence of an immunogenic microenvironment in SMARCB1-defective tumors [94], a dedicated phase II clinical trial evaluating the efficacy of nivolumab and ipilimumab SMARCB1-negative pediatric cancers has been launched (NCT04416568). Similarly, a phase II trial evaluating nivolumab plus ipilimumab in patients with SMARCB1-deficient renal malignancies is ongoing (NCT03274258). We can hope that these trials will bring interesting and more homogeneous results, to better understand the role of SMARCB1 in tumor immunogenicity and optimize the use of ICI in patients with SMARCB1-negative diseases.

#### 4.2.2. Anti-Angiogenic Agents and ICI

Combination with antiangiogenic therapeutics targeting VEGF or VEGF receptor (VEGFR) has been supported by pre-clinical evidence that VEGF inhibits T-cell development and may contribute to tumor-induced immune suppression [107]. Interestingly, a prolonged (24 weeks) objective response has been reported in the only patient with ES enrolled in the clinical trial evaluating the axitinib and pembrolizumab combination [108]. Intriguingly, and by contrast, no objective response was observed among the seven patients enrolled in the phase Ib/II trial evaluating combination of nivolumab with sunitinib [108,109]. Similarly, the combination of dasatinib (a multikinase inhibitor with an primarily anti-src family kinase activity) and ipilimumab did not show any efficacy in a patient with ES [110].

#### 4.2.3. Epigenetic Modulators and ICI Combinations

Multiple preclinical studies support that epigenetic modulators—and, notably, EZH2 inhibitors—have the potential to modulate tumor’s immunogenicity and anti-tumor immune response [111]. 

EZH2, a histone methyltransferase, is the core enzymatic subunit of the PRC2 complex. EZH2 catalyzes the methylation of lysine 27 on histone H3 (H3K27me3) at promoters and enhancers, leading to the epigenetic silencing of genes notably involved in cell fate and proliferation [112]. In normal cells, mSWI/SNF complexes oppose to PRC2 transcriptional repression. This epigenetic antagonism has been best characterized in SMARCB1-deficient MRT models where loss of SMARCB1 is associated with constitutive EZH2 activation and downstream oncogenic activity; in MRT preclinical models, EZH2 inhibition induced durable tumor regression and apoptosis [113,114,115]. These preclinical results have supported the development of Phase II basket multicenter clinical trial evaluating the EZH2 inhibitor tazemetostat in patients with SMARCB1-negative solid tumors (NCT02601950). On the basis of the results of ES cohort [43] which showed a response rate of 15% and some degree of tumor shrinkage or prolonged stabilization in more than half of the patients, the Food and Drug Administration (FDA) recently approved tazemetostat for the treatment of metastatic or locally advanced ES [116]. Beyond ES, EZH2 inhibitors have also shown efficacy in other SMARCB1-defective diseases in children, such as chordoma or ATRT, with 50% and 19% response rates, respectively [117].

Beyond their direct action through epigenetic antagonism, EZH2 inhibitors could also be used as immunomodulators, and the role of EZH2 in the modulation of the tumor immunogenicity has been extensively reviewed elsewhere [111,118]. In a nutshell, EZH2 inhibition can directly influence immunogenicity of cancer cells by (i): acting on the antigen presentation process: EZH2 expression contributes to the downregulation of Major Histocompatibility Complex (MHC)-I and MHC-II, which subsequently dampens the anti-tumor immune response [119,120]; (ii) activating the cell-autonomous innate immune signaling pathways: in chemo-resistant small cell lung cancer cells, inhibition of EZH2 favored the de-repression of the SPARCs, a subclass of ERVs, which triggered the activation of the type I IFN pathway [121]; (iii) rewiring the tumor immune microenvironment: EZH2 inhibition has been reported to result in the reactivation of TH1 cell-type cytokine expression, CXCL9 and CXCL10 and to increase infiltration of CD8+ T cells in human ovarian cancer models [122]. Through these various actions, EZH2 is both involved in the primary sensitivity to ICB, and in adaptive resistance mechanisms [120]. Multiple studies have reported that pharmacological inhibition of EZH2 is able to circumvent primary or acquired resistance to anti-PD-(L)1 resistance in several cancer types [120,123,124,125]. In line with these preclinical results, Gounder et al. reported a prolonged (>2 years) and exceptional abscopal response to radiotherapy in a patient with a SMARCB1-negative poorly differentiated chordoma who previously progressed on the EZH2 inhibitor tazemetostat [43]. Comparisons of tumor biopsy samples obtained prior to and during tazemetostat revealed a substantial increase in intra-tumoral and stromal infiltrates of CD8+ cytotoxic and FOXP3+ regulatory T cells, together with an enhanced expression of the PD-1 and LAG3 immune-checkpoint proteins on T cells. The ongoing multicenter phase II CAIRE clinical trial (NCT04705818, Table 4), which evaluates the association of anti-PD-L1, durvalumab and the EZH2 inhibitor tazemetostat in solid tumors including soft-tissue sarcoma, will shed light on the potential of this combination and results are eagerly awaited.

## 5. Perspectives and Future Directions

### 5.1. SMARCB1: A Role Which Is still Unclear in Modulating Tumor Immunogenicity

As described above, multiple converging preclinical data support that loss of SMARCB1 leads to a more immunogenic tumor microenvironment, at least in some tumor types. Though, several aspects are still unclear and deserve further investigation. 

First, SMARCB1-deficient tumors encompass multiple various neoplasms in different anatomic sites. In some of these, such as MRT, SMARCB1 is virtually the only genetic alteration, but in others, such as ES, the tumor genome is much more complex. Additionally, the mechanisms by which SMARCB1 loss may favor tumor immunogenicity can differ, as described above in MRT and RMC, suggesting some variability according to the cell of origin [66,73]. How these differences influence tumor immunogenicity is still unclear. Collaborative efforts, notably aiming at comparing the immune profiles of various diseases belonging to the “SMARCB1-deficient spectrum” of malignancies, would bring very useful insight on this question.

Similarly, even within a given SMARCB1-defective histological type, heterogeneity of immune infiltration has been reported, as illustrated by the various AT/RT methylation subgroups [72]. Deeper molecular characterization of larger series (including not only RNA sequencing but also methylome analysis, multiplex immunohistochemistry to define immune populations, and TLS assessment), should further shed light on this.

Finally, clinical results of ICI therapy efficacy in SMARCB1-defective sarcoma are still scarce due to the rarity of these diseases, and the heterogeneity of the available results in terms of patients’ characteristics (various histologic subtypes, treatment lines, etc.). Ongoing clinical trials will hopefully bring more homogeneous results and be associated with insightful translational studies which will eventually help in understanding the role of SMARCB1 defect in tumor immunogenicity.

### 5.2. Improving Patient Selection for Immune Checkpoint Inhibitors

Beyond SMARCB1, several important biomarkers should also be taken into account when selecting patients for immune checkpoint blockers targeting the PD-1/anti-PD-L1. For example, TLS were not assessed in studies that evaluated ICI in SMARCB1-deficient sarcoma so far. Additionally, transcriptional signatures which predict response to anti-PD-(L)1 therapies, such as an active IFN-γ signature, may further help identifying patients who are most likely to benefit from ICI therapy [126,127]. As RNA sequencing is becoming more and more widely used in sarcoma diagnostic and more generally in routine tumor profiling, this information should be exploited as well.

### 5.3. Other Immunotherapeutic Approaches

Other immune checkpoint inhibitors are currently being evaluated, such as relatlimab. This monoclonal antibody targeting lymphocyte activation gene-3 (LAG-3) is evaluated in combination with nivolumab in a phase II clinical trial enrolling patients with selected TLS-positive sarcoma (NCT04095208) (Table 4). Further, T-cell-based approaches (including CAR-T cells and genetically engineered TCRs) may represent a promising approach to target antigens highly or specifically expressed in certain types of SMARCB1-defective diseases. For example, preclinical in vivo studies in mice show that CAR-T cells directed against the B7/H3 checkpoint trigger a potent antitumor immune response with complete clearance of cerebral ATRT xenografts, when administered intracerebroventricularly or intratumorally [128].

## 6. Conclusions

Converging recent data support that SMARCB1 loss favors anti-tumor immunogenicity, at least in certain subtypes of SMARCB1-deficient sarcomas and carcinomas. Whether this can be generalized SMARCB1-deficient diseases remains an important and open question. Mechanisms by which SMARCB1 modulates tumor immunogenicity are diverse and potentially disease-specific, which suggests that clinical therapeutic approaches should be customized accordingly. Because SMARCB1-defective tumors are rare, clinical cases of response to immune checkpoint inhibitors are still anecdotal, but clinical trials are ongoing and series expanding rapidly. We can hope that the ongoing collaborative efforts on preclinical studies in SMARCB1-defective models and on the molecular characterization of SMARCB1-defective tumors, together with the growing results of several ongoing clinical studies, will allow to unravel the molecular determinants of response to immunotherapy in SMARCB1-deficient tumors. At a time where multiple forms of immune therapies are being development at an unprecedented speed, either as a monotherapy or in combination with other anti-tumor agents, this brings an unprecedented hope for the treatment of these deadly diseases and opens novel therapeutic avenues for patient’s benefit. 

## Figures and Tables

**Figure 1 biomedicines-10-00650-f001:**
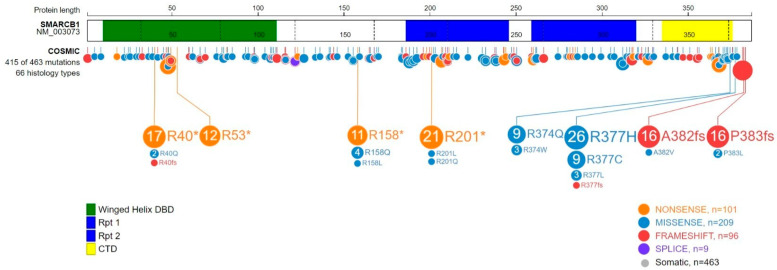
Schematic representation of SMARCB1 functional domains and summary of pathogenic somatic mutations (COSMIC). The SMARCB1 protein contains four functional domains: a winged helix domain DNA-binding domain, DBD (aa10-110), two highly conserved imperfect repeat domains, Rpt1 (aa186-248) and Rpt2 (aa259-319) and the highly conserved putative coiled-coil C-terminal α helix domain, CTD (aa335-375). Synonymous mutations have been excluded. Number of the most frequent pathogenic somatic mutations found in COSMIC are indicated.

**Figure 2 biomedicines-10-00650-f002:**
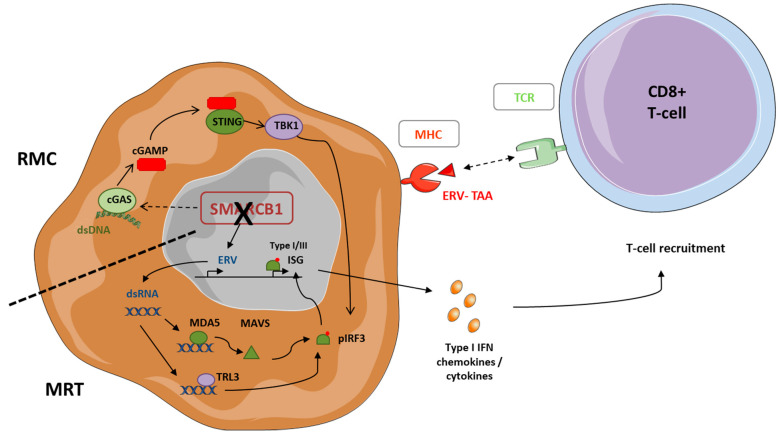
Interplay between SMARCB1 deficiency and immune modulation in SMARCB1-deficient MRT and RMC. Bottom right part of the tumor cell: in MRT, ERVs de-repression contributes to the accumulation of the cytosolic double-stranded RNA (dsRNA) which are recognized by the Toll-like receptor (TLR) 3 and MDA5 sensors. MDA5 binds to MAVS resulting in a signaling cascade which promotes the phosphorylation and nuclear translocation of IRF3, and subsequent induction of type I/III interferon-stimulated genes (ISG). This overall results in cytokine production which favors the recruitment of TILs. Aberrantly expressed ERV may also contribute to the development of an adaptive immune response through the production of tumor associated neoantigens (TAA). Upper left part of the tumor cell: in RMC and within a context of MYC-induced replication stress, dsDNA is released in the cytoplasm, which activates the cGAS/STING DNA-sensing pathway. The DNA sensor cGAS binds to dsDNA, which triggers the formation of cyclic G-AMP (cGAMP), and subsequently activates STING. cGAMP-bound STING recruits TBK1 and phosphotylates IRF3, which translocates to the nucleus where it triggers the expression of ISG.

**Table 1 biomedicines-10-00650-t001:** SMARCB1-deficient malignant neoplasms.

SMARCB1-Deficient Mesenchymal Malignant Tumors	SMARCB1-Deficient Non-Mesenchymal Malignant Tumors
Extrarenal malignant rhabdoid tumor	Atypical teratoid rhabdoid tumor
Epithelioid sarcoma	Cribriform neuroepithelial tumor
Poorly differentiated chordoma	Renal medullary carcinoma
Epithelioid MPNST ^1^	SMARCB1-deficient sinonasal carcinoma
Myoepithelial carcinoma	SMARCB1-deficient carcinoma of the GI tract
Myxoid extraskeletal chondrosarcoma	

^1^ malignant peripheral nerve sheath tumor.

**Table 2 biomedicines-10-00650-t002:** Results of clinical studies evaluating ICI in monotherapy in SMARCB1-deficient sarcoma.

Reference	NCT Identifier/ Trial Name	Study Design	Study Description	Number of Patients	Specific Histotype	Best Response	Duration of Best Response
Paoluzzi, 2016		Retrospective series	Nivolumab in relapsed metastatic/unresectable sarcomas	2	ES	1 PR	3.8 mth
						1 PD	
Blay, 2019	NCT03012620/ AcSé	Phase II	Pembrolizumab for patients with selected rare cancer types	1	MRT	PR	NA
Georger, 2020	NCT02541604/ iMATRIX	Phase I/II	Atezolizumab in children and young adults with refractory or relapsed solid tumors, with known or expected PD-L1 expression	3	MRT	PR	NA
Georger, 2020	NCT02332668	Phase I/II	Pembrolizumab in pediatric patients with PD-L1-positive, advanced, relapsed, or refractory solid tumor	2	MRT	1 PR	17.8 mth
						1 PD	
				1	ES	PR	11.8 mth
Forrest, 2020		Case report	Pembrolizumab	1	ES	SD	12 mth
			Nivolumab Pembrolizumab	1	PDC	PR	9 mth
				1	MRT	SD	15 wk

Abbreviations: ES: epithelioid sarcoma; MRT: malignant rhabdoid tumor; PDC: poorly differentiated chordoma; PR: partial response; PD: progressive disease; SD: stable disease; NA: not available.

**Table 3 biomedicines-10-00650-t003:** Results of clinical studies evaluating ICI therapy in combination in SMARCB1-deficient sarcoma.

Reference	NCT Identifier/Trial Name	Study Design	Study Description	Number of Patients	Specific Histotype	Best Response	Duration of Best Response
D’Angelo, 2018	NCT02500797/ Alliance A091401	Phase II	Nivolumab with or without ipilimumab treatment for metastatic sarcoma	1	ES	0	
Wilky, 2019	NCT02636725	Phase II	Axitinib + pembrolizumab in advanced sarcoma	1	ES	PR	24 wk
Martin-Broto, 2020	NCT03277924	Phase Ib/II	Nivolumab and sunitinib combination in advanced soft tissue sarcomas	7	ES	SD	17 mth
D’Angelo, 2017	NCT01643278	Phase Ib	Combined KIT and CTLA-4 Blockade in patients with Refractory GIST and other advanced Sarcomas: Ipilimumab + dasatinib	1	ES	SD	16 wk
Pecora, 2020		Case report	Nivolumab + ipilumab	1	ES	Complete response	NA

Abbreviations: ES: epithelioid sarcoma; PR: partial response; SD: stable disease; NA: not available.

**Table 4 biomedicines-10-00650-t004:** Selected ongoing immunotherapy-based clinical trials for SMARCB1-deficient sarcoma.

NCT Identifier/ Study Name	Drugs	Clinical Trial Phase	Population	Estimated Enrollment	Primary Completion Date	Location
NCT04741438/ RAR-Immune	Nivolumab + Ipilimumab	III	Metastatic or unresectable advanced sarcoma of rare subtype including ES and chordoma	96	February 2025	Centre Léon Bérard, Lyon, France
NCT04416568	Nivolumab + Ipilimumab	II	Relapsed or refractory INI1-negative cancers in children and young adults from 6 months to 30 years	45	October 2023	Dana-Farber Cancer Institute, United States, Massachusetts
NCT04705818/ CAIRE	Durvalumab + Tazemetostat	II	Distinct cohorts of solid tumors including soft-tissue sarcoma and metastatic solid tumor with positive interferon gamma signature and/or presence of TLS	173	October 2022	Multiple French Cancer Institute
NCT04095208/ CONGRATS	Nivolumab + Relatlimab	II	Advanced non-resectable/metastatic soft tissue sarcoma with high-level of tertiary lymphoid structures	67	March 2022	Multiple French Cancer Institute

Abbreviations: ES: epithelioid sarcoma; TLS: tertiary lymphoid structure.

## Data Availability

Not applicable.

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
