# Peer review of "Immunotherapy for SMARCB1-Deficient Sarcomas: Current Evidence and Future Developments"

_biomedicines, 2022, doi:10.3390/biomedicines10030650_

Round 1

Reviewer 1 Report

This manuscript is a very comprehensive, well-organized and, more important, a unique review of immunotherapy in SMARCB1-defective sarcomas. This treatment seems to be promising for these kinds of tumors. Unfortunately, the available data on immunotherapy’s efficacy is still scarce due to the rarity and the heterogeneity of these diseases. Therefore, I find it very valuable that the authors summarized them in one publication.

I suggest to accept the manuscript in present form

Author Response

We thank the reviewer for the largely positive comments.

Reviewer 2 Report

In the manuscript “Immunotherapy for SMARCB1-deficient sarcomas: current evidence and future developments” Ngo and Postel-Vinay provide a comprehensive review of existent preclinical and clinical data which suggest that immunotherapies may provide promising therapeutic strategies for SMARCB1-defective tumors.

The review is well written and discusses an important and timely subject.

A few suggestions:

  • Line 30-32. The authors could include the recently described DNA methylation-based sarcoma classifier as an additional molecular classification method (PMID: 33479225)

  • Line 59: Since the authors talk about the different BAF complexes would be good to explain the meaning of BAF: “mSWI/SNF remodelling complexes, also known as BRG1/BRM-associated factor (BAF) complexes, exist in three forms…”

  • Line 73-74: “SMARCB1 was first identified as a binding partner of the human immunodefi- ciency virus-type 1 (HIV-1) integrase through two its highly conserved imperfect repeat domains, and thereafter named integrase interactor 1, INI1”

The sentence above contains a typo or is not well constructed. Please correct.

  • Line 94-99. This sentence is too long. Consider to break in two for clarity.

  • Line 132: First, the mechanisms leading to SMARCB1 inactivation are more diverse..”

  • Line 193: “Using immunocompetent mouse models..”

  • Line 199: “..also reported that de-rexpressed ERVs..” Should be de-repressed?

  • Line 284: Typo: “FDA-approved immunohistochemistry companion diagnostic tests” company?

  • Line 309: “..suggesting that T-cell exhaustion that may contribute to immune 309 escape in these diseases [64]. “ I think the second “that” should be deleted.

  • Line 386: Add “is” in the following sentence. “This epigenetic antagonism has been best characterized in SMARCB1-deficient MRT models where loss of SMARCB1 is associated with constitutive EZH2 activation”

Author Response

Response to Reviewer 2 Comments

Point 1: Line 30-32. The authors could include the recently described DNA methylation-based sarcoma classifier as an additional molecular classification method (PMID: 33479225)

Response 1: We thank the reviewer for this very relevant suggestion and have updated the manuscript accordinngly. It now reads: “More recently, other classifiers have emerged, notably based on DNA methylation profiles [3 - PMID: 33479225], or on digital pathology associated to artificial intelligence / deep learning analyses, which all represents valuable additional tools for sarcoma classification [4 – PMID 34139273].

Beyond genetic abnormalities, several sarcoma subtypes are driven, at least in part, by epigenetic dysregulation. […]”

Point 2: Line 59: Since the authors talk about the different BAF complexes would be good to explain the meaning of BAF: “mSWI/SNF remodelling complexes, also known as BRG1/BRM-associated factor (BAF) complexes, exist in three forms…”

Response 2: The manuscript has now been corrected as suggested by the reviewer.

Point 3: Line 73-74: “SMARCB1 was first identified as a binding partner of the human immunodefi- ciency virus-type 1 (HIV-1) integrase through two its highly conserved imperfect repeat domains, and thereafter named integrase interactor 1, INI1” The sentence above contains a typo or is not well constructed. Please correct.

Response 3: The manuscript has been revised accordingly and now reads: “SMARCB1 was first identified as a binding partner of the human immunodeficiency virus-type 1 (HIV-1) integrase (to which it binds through its two highly conserved imperfect repeat domains, Rpt1 and Rpt2), and was thereafter named integrase interactor 1, INI1.

Point 4: Line 94-99. This sentence is too long. Consider to break in two for clarity.

Response 4: The manuscript has been updated and now reads: “SMARCB1 also commonly acts as a tumor suppressor by transcriptionally regulating the cell cycle, proliferation, and differentiation. For example, SMARCB1 has been shown to regulate the activation of CyclinD1/CDK4 signaling, to repress RB target genes, and to participate to the regulation of c-MYC-associated transcriptional programs.”

Point 5: Line 132: First, the mechanisms leading to SMARCB1 inactivation are more diverse..”

Response 5: The manuscript has been corrected as suggested by the reviewer.

Point 6: Line 193: “Using immunocompetent mouse models..”

Response 6: The manuscript has been corrected as suggested by the reviewer.

Point 7: Line 199: “..also reported that de-rexpressed ERVs..” Should be de-repressed?

Response 7: We apologize for the typo and indeed meant de-repressed ERVs. The manuscript has been corrected accordingly and now reads: “Interestingly, recent studies have also reported that de-repressed ERVs could generate tumor specific antigens presented at the tumor cell surface by MHC-I molecules and recognized by specific T-cells”.

Point 8: Line 284: Typo: “FDA-approved immunohistochemistry companion diagnostic tests” company?

Response 8: We thank the reviewer for this suggestion: the companies have now been added and the paragraph accordingly reads: “. High PD-L1 expression, using FDA-approved immunohistochemistry companion diagnostic tests (e.g. 22C3 pharmDx on Dako Omnis, Agilent, for pembrolizumab, MSD; or SP263 on Ventana for atezolizumab, Roche) is currently the most robust predictive biomarker of response to anti-PD-(L)1 agents, and is routinely used in patients with advanced NSCLC, bladder cancer, and head and neck squamous cell carcinoma”.

Point 9: Line 309: “..suggesting that T-cell exhaustion that may contribute to immune escape in these diseases [64]. “ I think the second “that” should be deleted.

Response 9: The manuscript has been corrected as suggested by the reviewer.

Point 10: Line 386: Add “is” in the following sentence. “This epigenetic antagonism has been best characterized in SMARCB1-deficient MRT models where loss of SMARCB1 is associated with constitutive EZH2 activation”

Response 10: The manuscript has been corrected as suggested by the reviewer.
